# Food security and nutritional vulnerability in Comoros: The impact of Russia-Ukraine conflict

Estefanía Custodio[1,2], Maria Priscila Ramos[3,4], Sofía Jimenez[5,6], Francis Mulangu[7], Nicolas Depetris-Chauvin[8]*

1 CIBER Infectious diseases, Carlos III Health Institute (ISCIII), Madrid, Spain, 2 National Centre for Tropical Diseases (CNMT), Carlos III Health Institute (ISCIII), Madrid, Spain, 3 Departamento de Economía, Facultad de Ciencias Económicas, Universidad de Buenos Aires, Buenos Aires, Argentina, 4 CONICET-Universidad de Buenos Aires, Instituto Interdisciplinario de Economía Política de Buenos Aires, Buenos Aires, Argentina, 5 Economic Analysis Department, School of Business and Economics, University of Zaragoza, Zaragoza, Spain, 6 Aragonese Agrifood Institute, Zaragoza, Spain, 7 World Bank, Poverty and Equity Global Practice, Antananarivo, Madagascar, 8 HES-SO/Haute Ecole de Gestion de Genève, University of Applied Sciences and Arts of Western Switzerland, Geneva, Switzerland

* nicolas.depetris-chauvin@hesge.ch

**Data Availability Statement:** The data of the Comoros National Household survey is available upon request from the Nataional Statistics Institute of Comoros (INSEED), at INSEED's website https://

## Abstract

The Sustainable Development Goal 2 aims to eradicate hunger. However, many small island developing states (SIDS) face challenges in achieving this goal due to their reliance on the global market and susceptibility to crises. This paper focuses on Comoros, one of the least studied SIDS located in the Indian Ocean, to examine its food security and nutritional vulnerability. The findings reveal that Comoros experiences limited access to sufficient and nutritious diets, which are excessive in fat and deficient in key micronutrients. This situation is similar to other SIDS in the Pacific and the Caribbean, which also exhibit the convergence of the malnutrition triple burden. This paper further assesses the impact of changing food prices during the initial year of the Russia-Ukraine conflict on food security and nutrition in Comoros, serving as a case study for SIDS. Using microsimulations, we estimate the effects on food access, sufficiency, and adequacy at the household level. The results indicate a decline in food access and purchasing power for all households, particularly the poorer and rural. Additionally, food sufficiency has markedly decreased, leading to an additional 5,000 households (3.3% of all households) experiencing inadequate daily calorie intake. The study also highlights a reduction in the consumption of organ meats, fruits, and vegetables, resulting in decreased daily iron and folate intakes. This is particularly concerning given the high prevalence of anemia among women in Comoros and other SIDS. To address these challenges, policies promoting the consumption of fresh, nutrient-rich foods with low fat content are crucial to mitigate the malnutrition triple burden and curb the increasing trend of non-communicable diseases in these countries.

www.inseed-comores.org/category/enquetes/ehcvm-2020/. The Comoros Nutrients Composition Table elaborated by the authors has been attached to this resubmission. All other data and codes are attached as a zip file.

**Funding:** The author(s) received no specific funding for this work.

**Competing interests:** No authors have competing interests

## 1. Introduction

The United Nations (UN) Development Programme acknowledges the significance of Sustainable Development Goal 2 (SDG2), which aims to eliminate hunger and all forms of malnutrition. However, a substantial number of small island developing states (SIDS) encounter difficulties in achieving this goal. The UN designates 58 countries and territories as SIDS, with 38 being full UN members. These SIDS are widely acknowledged as a distinctive category for sustainable development due to the unique array of challenges and vulnerabilities they confront. Predominantly located in the Caribbean and the Pacific, eight are situated in the Atlantic, Indian Ocean, and South China Sea region (AIS). Despite their heterogeneity, most share common traits that render them particularly susceptible to food security and nutrition (FS&N) hazards. Certain characteristics stem from their specific geography, such as insularity or limited arable land, and heightened exposure to the impacts of climate variability and natural disasters. Other traits are of an economic and social nature, encompassing rapid population growth and urbanization, constrained employment and export opportunities, and a high import dependency that renders these nations especially vulnerable to global economic shifts [1, 2].

These challenges have recently gained prominence with disruptions in global food supply chains caused by the Covid-19 pandemic [3, 4] and the global food price fluctuations resulting from the Russia-Ukraine conflict that started in February 2022 [5, 6]. According to [7], this conflict added 2% to global inflation in 2022 and 1% in 2023. Moreover, countries in Sub-Saharan Africa, such as Tanzania or Kenya, have especially experienced high inflation rates due to this conflict, which has impacted strongly on the welfare of its population [8, 9].

The Russia-Ukraine conflict has instigated a crisis in food supply chains through two primary channels that contribute to the escalation of food prices. Firstly, it directly reduces the production of agricultural commodities worldwide. Secondly, it has driven up transportation costs for food and fertilizers [8, 10, 11]. Although food access declined globally, the most severely affected were the less developed and small importing countries, particularly in Africa and the Middle East [5, 12, 13]. Their food security was jeopardized [14–17], especially for women of reproductive age, children in their early years, and the urban poor overall [18]. In the extreme scenario of no wheat exports from Ukraine and Russia, it is estimated that the daily per capita calorie intake would decrease by 6% in Sub-Saharan Africa and by 14% in the Middle East and North Africa [19]. For the SIDS, highly dependent on imported goods and deeply integrated into global markets, these market disruptions can have an even greater impact on their food security and nutritional patterns [20]. They already have limited and weakened local food systems, due to accelerated rural emigration and urbanization, and are constrained by an almost nonexistent food storage infrastructure [3]. Consequently, the composition of diets in most SIDS has shifted from domestic staples, fruits, and vegetables to heavily caloric processed foods, meat, and dairy products [21–23], resulting in increasing rates of adult obesity and micronutrient deficiencies, such as anemia, especially among women of reproductive age and children under five [3]. These nutritional disorders and imbalances are closely linked to non-communicable diseases (NCDs), which are the primary cause of illness and mortality in most SIDS [24].

The SIDS in the AIS region exhibit economic heterogeneity, with four of them classified as middle-high or high-income countries (Singapore, Maldives, Mauritius, and Seychelles), while the remaining four fall into the middle-low or low-income category (Cabo Verde, Guinea Bissau, Sao Tome e Principe, and Comoros). The average SDG2 score for all the AIS SIDS is relatively high, but it obscures the food insecurity and nutritional challenges faced by the latter

group of islands and archipelagos Low-income SIDS countries have been less frequently studied compared to their counterparts in the Pacific and Caribbean SIDS [1].

Comoros belongs to the least developed countries category according to the United Nations Department of Economic and Social Affairs [25] and it has been hit by multiple of the aforementioned recent shocks, including Cyclone Kenneth and COVID-19 pandemic [26]. It grapples with a high poverty rate (it was 39.8% in 2022 and exceeded the 12% average for lower-middle-income countries) exacerbated by the COVID-19 pandemic and subsequent lockdowns [27]. Besides, Comoros faces ongoing issues related to food and nutrition security, characterized by a high stunting rate among children under five (31.1% in 2020), surpassing the African average (30.7%). Comoros is ranked as the country with the highest stunting rate among all SIDS [28]. The country also shows elevated rates of obesity (14% among adult women and 4% among adult men) and anemia (34% among women of reproductive age). Comoros' nutritional profile aligns with that of other SIDS in the AIS region, the Caribbean, and the Pacific [21–23] although there is scarce information on the patterns of food consumption and dietary patterns of its population.

Like other SIDS, Comoros heavily depends on imports of cereals and oilseeds from Ukraine and Russia. The impact of the Russia-Ukraine conflict became noticeable in Comoros around June 2022, coinciding with a rise in inflation (e.g., the annual inflation rate, which stayed below 7% before the conflict, surged to 11.4% in June 2022 and reached 20.6% by December 2022). In 2023, with the conflict persisting, the inflation rate remained high, averaging 17.4%. The policy response to this situation was mixed. On the one hand, the government raised domestic fuel and electricity prices, which increased by 44% in June 2022, to face the higher import costs. On the other hand, it increased subsidies to the monopolist rice importer ONICOR (Office National d'Importation de Commercialisation du Riz) and launched a subsidy for bakeries to avoid hikes in the prices of rice and flour-based products [22].

The case study of Comoros can provide valuable insights into addressing common challenges in achieving SDG2 for lower-income SIDS, especially when global events such as the Russia-Ukraine war impact food production and prices, compromising food access and altering consumption patterns, thereby affecting calories and nutrient intake [9, 6].

This paper aims to describe the food security and nutrition (FS&N) situation, with special emphasis on food consumption and dietary adequacy in Comoros, an understudied SIDS. It also evaluates the impact of food price shocks one year after the Russian invasion of Ukraine on FS&N indicators, including calories, macronutrients, and micronutrients intake. This case study serves as an instructive model for other low- and middle-income SIDS. The existing empirical literature on the FS&N implications of the Russia-Ukraine conflict in SIDS, among other net-importing countries, focuses on estimations of agricultural trade dependence and vulnerability of local food systems in SIDS [2, 29]. Furthermore, these analyses mainly consider changes in food expenditure shares, calories, and protein intake solely from wheat consumption [16, 19, 30]. To the best of our knowledge, no published paper extends the analysis of the Russia-Ukraine conflict to a comprehensive examination of FS&N based on households' diets in SIDS. Therefore, this paper fills a crucial gap in the literature by examining the impact on nutrients, providing essential information for targeted policy recommendations.

The remainder of the paper is organized as follows. The next section outlines the methodology used to construct the microsimulations and prepare the data. Section 3 presents and discusses the results of the analysis regarding the FS&N situation in Comoros and the impact of the Russia-Ukraine conflict on FS&N indicators at the household level. Section 4 concludes by providing recommendations for future nutrition policies in Comoros, many of them also applicable to similar SIDS.

## 2. Methodology

### 2.1. Microsimulation approach for food security and nutrition analysis

To assess the impact of changes in food prices on the FS&N dimensions–specifically, food access, food sufficiency, and food adequacy–at the household level in Comoros, we conducted simulations using the FS&N approach employed by [31–33]. However, our study extends the analysis to include a comprehensive examination of micronutrients, a facet not covered in those previous works. Subsequently, we performed non-parametric regressions to gauge the distributional effects of FS&N indicators across per capita expenditure for all national households, distinguishing between rural and urban locations.

Fig 1 illustrates the general methodology we followed. We first establish the input data, encompassing household survey data, food composition tables, and food prices. We then applied data treatments and introduced shocks, specifically the changes in real food prices during the initial year of the Russia-Ukraine conflict. Subsequently, microsimulations were conducted on FS&N dimensions. To account for potential non-linearities, we employed non-parametric smoothing regressions. Finally, the results elucidate the changes in the FS&N situation of households when subjected to an exogenous shock impacting their diets.

### 2.2. Microdata requirement and treatment

To conduct the FS&N analysis, we require household survey data that includes consumption details for each food item in both monetary and physical units, along with a Nutrients Composition Table (NCT) tailored to the population under study.

Comoros' Institute National de la Statistique et des Etudes Economique et Démographiques (INSEED) recently conducted a household survey (Enquête Harmonisée sur les Conditions de Vie des Ménages 2018/2019 –EHCVM 2020) involving 5,624 households. The survey captures the consumption of 148 food items, providing data on food consumption, expenditures, and household characteristics, including location (rural/urban) and family composition (household size, gender of household head, number of children, etc.), essential for calculating food expenditure and consumption indicators. Additionally, to construct the food consumption

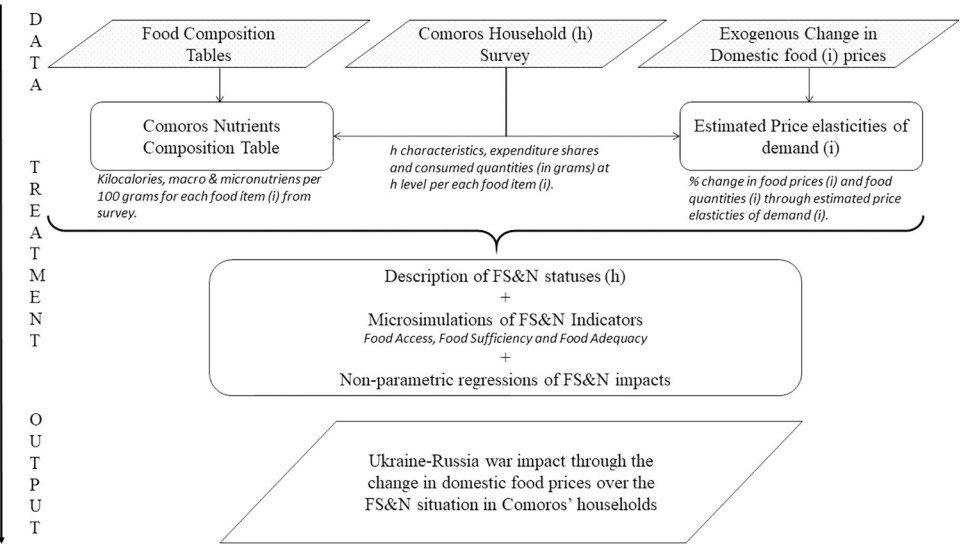

**Fig 1. Methodological approach: General scheme.** Source: own elaboration.

variables, quantities consumed are standardized into grams, given the diverse units used (e.g., liters of fresh milk, bowls of cornflakes, etc.).

A Nutrients Composition Table (NCT) furnishes details on the energy and content of macro and micronutrients of the food items reported in the survey, by matching them with the food items included in the corresponding Food Composition Table (FCT). The FCTs are specific to countries or regions and contain biochemical information on calories and nutrients on food items commonly consumed by the population. There is no specific FCT for Comoros thus, we used the biochemical data of food items described in five different FCTs. The first and main FCT used was the Food Composition Table for Western Africa (used to match 72% of the food items), followed by Kenya Food Composition Table (14% of food items), USDA Food Database (7% of food items), Spanish Food Composition Database (3%) and Brazilian Food Composition Table (0.7%). We then constructed the Comoros NCT by matching the food items gathered in the food consumption section of the EHCVM with the corresponding food item included in one of the five FCTs described, and computed the calories and nutrients estimates by applying the corresponding adjustment factors (refuse, yield, and retention factors) to consider potential loss of nutrients due to processing and cooking [34].

To analyze domestic real food price changes in Comoros during the initial year after the commencement of the Russia-Ukraine conflict, we utilized INSEED's price index for 152 food items (8-digit good code) spanning from February 2022 to February 2023. We also considered the general CPI to calculate the inflation rate during the same period (19.9%). The 152 food codes were matched and aggregated (weighted) into the 148 food items in the consumption module of the EHCVM.

The price-demand elasticities for the principal food items within each food group are directly taken from the World Bank estimates. This allowed us to convert the changes in food prices (after deducting national inflation) into the impact on consumed food quantities, subsequently simulating the consequences on food sufficiency and adequacy indicators at the household level (refer to these estimates in the S1 File–Table A.1).

## 2.3. FS&N indicators

Presently, no single indicator comprehensively captures all dimensions of food security. Instead, a set of appropriate indicators is necessary to delineate each dimension in an integrated manner [35]. In alignment with the FS&N literature and constrained by the available data in the Comoros survey, our focus is on assessing the impact on the dimensions of food access and food consumption, employing the methodology outlined by [32]. For food access, we gauge the alteration in food purchasing power. Food consumption is further dissected into two sub-dimensions: food sufficiency, tracking changes in daily calorie intake per capita, and food adequacy, assessing changes in daily macronutrients and micronutrients per capita intake at the household level.

Given exogenous domestic price changes ($\Delta p$) for each food item (i) and given the households' (h) food expenditure shares ($\theta$) of each food item (i), we compute the food access (FA) impact. The food access impact is the change in the per capita purchasing power of food at the household level (Eq 1), that says food access deteriorates when food prices increase (inverse relation). It shows the change in the accessibility of food based on its affordability.

$$\Delta FA_h = -\sum_{i=1}^{n} \left( \theta_{i,h} \cdot \Delta p_i \right) \tag{1}$$

For the food consumption impact, that is to measure both the change in food sufficiency and food adequacy indicators for Comorian households, we need the change in food quantities

consumed per food item per day ($\Delta q$) and not only the $\Delta p$ of each food item. So, using the estimates of price-demand elasticities ($\varepsilon$) we translate the changes in $\Delta p$ into $\Delta q$.

Food sufficiency represents the daily calorie intake per person, which is generally compared to a minimum requirement of energy per adult equivalent (2,250 kcal per day) suggested by FAO & WHO. Thus, according to the grams consumed in each household of each food item ($i$) and knowing the kilocalories contribution per gram of each food item ($\varphi$) from the Comoros' NCT, we calculate the change in total kilocalories consumed per day per capita in a household based on each household's diet. This indicator is the change in Dietary Energy Consumption (DEC), calculated as in expression (2).

$$\Delta DEC_h = \sum_{i=1}^{n} (\varphi_i \cdot \Delta q_{i,h}) \tag{2}$$

Food adequacy introduces a nutritional (NU) dimension of food consumption. It covers the calculation of the daily macronutrients (proteins, fats, carbohydrates) and micronutrients (calcium, iron, zinc, folate, vitamins C, B1, B2, A (RAE)) intakes per capita. To assess the diet in terms of those nutrients (j), we follow the methodology described by the World Bank and the FAO [36, 37]. The NCT provides the nutritional contribution ($\gamma$) of each food item per gram to calculate the daily nutrients (j) intake per capita for each household. Macronutrients contributions per gram of each food item are expressed in kilocalories, while the units of micronutrients contributions vary by micronutrient; e.g. calcium is expressed in milligrams and folate in micrograms per gram of edible portion of each food item. Thus, by summing up the change in daily nutrients intake per capita for every food item consumed in the household, we obtain the change in daily macro and micronutrients intakes per capita (per type of nutrient) at the household level (Eq 3).

$$\Delta NU_{j,h} = \sum_{i=1}^{n} (\gamma_{j,i} \cdot \Delta q_{i,h}) \tag{3}$$

Non-parametric regressions are conducted on these microsimulation results, across percentiles of households based on their per capita expenditure.

## 3. Results

### 3.1. Current foods security and nutrition situation in Comoros: Descriptive statistics

In Table 1 we present the descriptive statistics of the food access and food consumption (sufficiency and adequacy) indicators for national, rural, and urban households in the Comoros island state.

The average household size in Comoros is 5.2 family members, with an average food expenditure share of 55%, considered medium [38], and closely aligning with Kenya's 56% [32]. The national average of daily energy consumed per capita stands at 2,287 kcal, while the median is lower at 2,042 kcal, indicating a right-skewed distribution. This positions Comoros in the lower range compared to other SIDS, as most hover around 3,000 Kcal of dietary energy in the food supply, such as Trinidad and Tobago, Nauru, and Vanuatu. However, others like Haiti or Guinea Bissau have lower values (2,091 kcal/capita/day and 2,292 kcal/capita/day, respectively) [39]. Merely 42% of all Comorian households meet the minimum recommendation of 2,250 kcal consumed per day per capita [40], with this percentage lower in rural areas (38%) and higher in urban ones (46%). The caloric intake shares provided by fats, proteins, and carbohydrates within the total caloric intake are, on average, 35%, 14%, and 48%, respectively. These values reveal an imbalance, with calories from fats outweighing those from carbohydrates. According to WHO and FAO recommendations for a balanced diet (refer to Table A.2. in the

**Table 1. Descriptive statistics of the population's daily food access and food consumption (sufficiency and adequacy).**

| | | | National | | | Rural | | | Urban | | |
|---|---|---|---|---|---|---|---|---|---|---|---|
| | | | N = | 151,825 | | | 66% | | | 34% | |
| | | | Mean | Median | Std. Dev. | Mean | Median | Std. Dev. | Mean | Median | Std. Dev. |
| | | HH size | 5.24 | 5 | 2.56 | 5.47 | 5 | 2.62 | 4.81 | 5 | 2.38 |
| | | Total Expenditure (per capita per day in KMF) | 2,302 | 1,761 | 1,946 | 2,044 | 1,599 | 1,639 | 2,796 | 2,183 | 2,350 |
| **Food Access** | | | | | | | | | | | |
| | | Food Expenditure (share) | 0.55 | 0.56 | 0.15 | 0.55 | 0.57 | 0.16 | 0.55 | 0.55 | 0.15 |
| | | HCES-DDS | 12 | 12 | 2 | 11 | 12 | 2 | 12 | 12 | 2 |
| **Food Consumption** | | | | | | | | | | | |
| Food sufficiency | | Energy intake* | 2,287 | 2,042 | 1,140 | 2,209 | 1,964 | 1,090 | 2,437 | 2,185 | 1,216 |
| Food Adequacy | Macronutrients | Protein intake* | 314 | 278 | 159 | 296 | 264 | 146 | 348 | 311 | 176 |
| | | Protein (share) | 0.14 | 0.14 | 0.04 | 0.14 | 0.13 | 0.04 | 0.15 | 0.14 | 0.04 |
| | | Fat intake* | 822 | 705 | 512 | 797 | 687 | 491 | 869 | 740 | 547 |
| | | Fat kcal (share) | 0.35 | 0.35 | 0.1 | 0.35 | 0.35 | 0.1 | 0.35 | 0.35 | 0.1 |
| | | Carbohydrates intake* | 1,076 | 937 | 575 | 1,040 | 913 | 554 | 1,145 | 999 | 606 |
| | | Carbohydrates (share) | 0.48 | 0.47 | 0.1 | 0.48 | 0.47 | 0.1 | 0.48 | 0.47 | 0.1 |
| | Micronutrients | Calcium intake* | 476 | 383 | 348 | 432 | 356 | 300 | 560 | 450 | 412 |
| | | Iron intake* | 14.36 | 12.59 | 7.76 | 13.89 | 12.16 | 7.47 | 15.28 | 13.48 | 8.21 |
| | | Zinc intake* | 7.76 | 6.76 | 4.23 | 7.4 | 6.46 | 3.86 | 8.47 | 7.29 | 4.79 |
| | | Folate intake* | 174 | 132 | 140 | 164 | 125 | 130 | 192 | 146 | 155 |
| | | Vit C intake* | 104 | 80 | 91 | 101 | 77 | 91 | 111 | 85 | 91 |
| | | Vit B1 intake* | 0.74 | 0.65 | 0.4 | 0.71 | 0.62 | 0.37 | 0.81 | 0.71 | 0.45 |
| | | Vit B2 intake* | 0.91 | 0.75 | 0.6 | 0.84 | 0.7 | 0.54 | 1.03 | 0.86 | 0.69 |
| | | Vit A RAE intake* | 518 | 248 | 1,097 | 479 | 231 | 1,010 | 592 | 286 | 1,244 |

Notes

*Energy and macronutrients intake in kcal per day per capita. Calcium, iron, zinc, vitamins B1, B2 and C are in milligrams per day per capita. Folate and vitamins A are in micrograms per day per capita. In order to compute the mean intake of macronutrients in grams, the quantities in kcal have to be divided by the corresponding Atwater factors (9 for fats and 4 for carbohydrates and proteins).

Source: own elaboration.

S1 File), the caloric share provided by carbohydrates should range between 55% and 75%, and calories from fats should not exceed 30% [41]. We estimate that in 71% of households at the national level, the consumption of calories from fats exceeds the recommended levels. The mean amount of fat consumed in Comoros is 91 grams per day per capita, which closely aligns with the mean fat intake in Fiji and Guam Island, countries with some of the highest obesity rates globally [42]. Finally, the estimates of daily per capita intake of micronutrients, excluding iron and vitamin C, fall below the estimated average requirements (EAR) for an adult male [43].

This scenario highlights a concerning situation in terms of FS&N in Comoros, characterized by widespread deficiencies in micronutrient intakes and unbalanced diets, particularly in terms of fat consumption. Although this survey lacks anthropometric measures of households' members, [28] asserts that overweight and obesity are prevalent, showing upward trends among adults and adolescents. These findings align with the triple burden of malnutrition observed in other SIDS, involving the coexistence of overweight and obesity, childhood stunting, and micronutrient deficiencies, especially iron deficiency in women [3].

In Table 2, we illustrate the contribution of different food groups in terms of daily per capita intake of energy, macronutrients, and micronutrients. Our observations reveal that in

**Table 2. Energy and nutrients contributions based on the national average daily diet per capita in Comoros.**

| | | 1 Cereals | 2 White Roots & Tubers | 3 Vit. A Rich Vegetables & Tubers | 4 Dark Green Leafy Vegetables | 5 Other Vegetables | 6 Vit. A Rich Fruits | 7 Other Fruits | 8 Organ Meat | 9 Flesh Meats | 10 Eggs | 11 Fish & Seafood | 12 Legumes, Nuts & Seeds | 13 Milk & Milk Products | 14 Oils & Fats | 15 Sweets | 16 Spices, Condiments, Beverages |
|---|---|---|---|---|---|---|---|---|---|---|---|---|---|---|---|---|---|
| Energy | | 16% | 8% | 3% | 1% | 1% | 4% | 11% | 6% | 13% | 1% | 4% | 3% | 4% | 15% | 10% | 1% |
| Macronutrients | Fat | 1% | 1% | 0% | 0% | 0% | 0% | 18% | 6% | 18% | 2% | 3% | 2% | 5% | 40% | 3% | 0% |
| | Pro. | 8% | 2% | 1% | 2% | 0% | 1% | 3% | 19% | 32% | 2% | 18% | 4% | 5% | 0% | 1% | 0% |
| | Car. | 34% | 16% | 5% | 1% | 1% | 8% | 6% | 1% | 0% | 0% | 0% | 3% | 3% | 0% | 21% | 2% |
| Micronutrients | Calc. | 8% | 4% | 4% | 14% | 1% | 4% | 5% | 2% | 3% | 2% | 4% | 3% | 32% | 0% | 8% | 5% |
| | Iron | 9% | 5% | 5% | 6% | 2% | 3% | 8% | 38% | 5% | 2% | 4% | 5% | 1% | 0% | 4% | 4% |
| | Zinc | 12% | 7% | 3% | 3% | 1% | 1% | 6% | 28% | 20% | 2% | 4% | 4% | 6% | 0% | 2% | 1% |
| | Folate | 7% | 3% | 3% | 5% | 1% | 10% | 2% | 56% | 1% | 2% | 1% | 6% | 1% | 0% | 1% | 1% |
| | Vit. C | 0% | 13% | 15% | 5% | 5% | 38% | 11% | 9% | 0% | 0% | 0% | 0% | 1% | 0% | 1% | 2% |
| | Vit. B1 | 12% | 6% | 5% | 5% | 2% | 5% | 4% | 21% | 9% | 1% | 9% | 10% | 5% | 0% | 5% | 1% |
| | Vit. B2 | 2% | 2% | 1% | 4% | 1% | 2% | 2% | 59% | 10% | 2% | 3% | 1% | 8% | 0% | 4% | 0% |
| | Vit. A RAE | 0% | 1% | 2% | 1% | 0% | 3% | 0% | 92% | 0% | 0% | 0% | 0% | 1% | 0% | 0% | 0% |

HCES-DDS food groups*

*Shaded in grey the food groups that contribute to the daily intake per capita of each component by 10% or more

Source: own elaboration.

Comorian diets, the primary sources of energy include cereals (16%), oils & fats (15%), flesh meats (13%), other fruits (11%), and sweets (10%). Conversely, the main sources of fats are oils and fats (40%), followed by the other fruits group (18%) and flesh meats (18%). This category of other fruits encompasses coconut and avocado, both fruits with high fat content commonly consumed in Comoros. Dry and fresh coconut, especially, play a prominent role in Comorian recipes, with a daily mean consumption of 93 grams per capita.

Proteins primarily derive from organ meats, flesh meats, and fish (33%, 20%, and 19% respectively). The consumption of organ meats is prevalent in small island nations due to isolation, which compels them to minimize waste and make the most of available meat by consuming organs rich in valuable nutrients, otherwise scarce in the limited nutritional options on the island. However, these are pricey products mostly accessible to households with high expenditures. Within the population studied in Comoros, those in the bottom 20 percent of the income distribution consume an average of 50 grams per capita of organ meats, while households in the top 20 percent consume over 70 grams per capita. Concerning flesh meat, chicken is the most frequently consumed, with a significant portion reported to be purchased as frozen meat due to the prevalence of imported meat on the island.

Lastly, the primary sources of carbohydrates are cereals (34%), followed by sweets (21%) and white roots and tubers (16%). The substantial share of carbohydrate calories provided by the sweets group is a concern and may be associated with the high rates of overweight and obesity in the country.

Regarding micronutrients, covering mineral and vitamin intakes, the main source of calcium is milk and milk products (32%), followed by the group of dark green leafy vegetables (14%). Zinc is primarily provided by flesh and organ meats (20% and 28% respectively) and cereals (12%), while the main sources of iron are organ meats (38%) followed by cereals (9%). The group of flesh meats contributes only 5% of the total iron in the Comorian diet, possibly because chicken, with lower iron content, is the main meat consumed. Organ meats are exceptionally rich in all micronutrients and serve as the main providers of most minerals and vitamins. One noteworthy source of vitamin C (38%) is the group of vitamin A rich fruits that mostly include papaya and mango, primarily consumed by wealthier households (consuming more than 150 grams per capita per day) compared to poorer households consuming less than 100 daily grams per capita.

## 3.2. Food security & nutrition impact in Comoros after one year of the Russia-Ukraine conflict: Microsimulation results

Considering the dietary patterns previously identified for households in Comoros, we examine how the change in food prices during the first year since the onset of the Russia-Ukraine conflict has influenced food access and food consumption, encompassing food sufficiency and adequacy.

**3.2.1. Change in food domestic prices and food quantities consumed.** In Fig 2, we show that domestic prices primarily rose in real terms for food groups rich in nutrients and low in calories, such as fish and seafood, cereals, organ meat, legumes, and various fruits and vegetables. Consequently, we observe a decline in the consumption of these groups, ranging from -18% and -17% for the legumes & nuts group, and the organ meat, respectively, to -1% for dark green leafy vegetables.

Conversely, prices decreased, and consumption increased for vitamin A-rich vegetables and tubers, flesh meats, eggs, oils and fats, sweets, and the spices and beverages group, which includes soda and alcoholic drinks. The heightened intake of these food groups (excluding the vitamin A-rich vegetables and tubers group) in a population already experiencing excessive fat

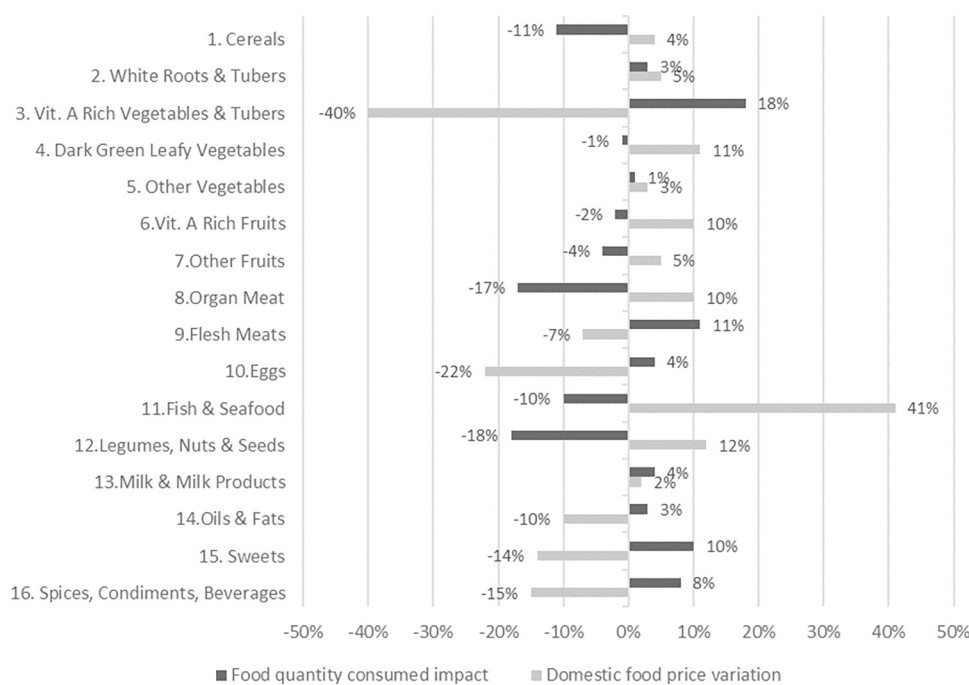

**Fig 2. Russia-Ukraine conflict impact–Average change in food prices and food quantities consumed by food group in Comoros.** Source: own elaboration based on food price indexes at 8-digit level (Feb22-Feb23) and the general CP Index from INSEED-Comoros.

intake and micronutrient deficiencies is concerning. This situation elevates the risk of NCDs and has been negatively associated with overweight and obesity in other SIDS [44, 45]. Moreover, the increased intake of sweets may be related to the subsidies given to bakeries by the Government to avoid the hike in the prices of rice and flour-based products [26].

**3.2.2. Impact in food access, food sufficiency and food adequacy.** Figs 3 and 4 present the change in the FS&N indicators over the study period across percentiles of per capita expenditure for all households included in the Comoros survey.

Overall, there is a decline in **food access** (Fig 3, panel A), indicated by the negative change in the **food purchasing power** of households. This decline is comparatively more pronounced,

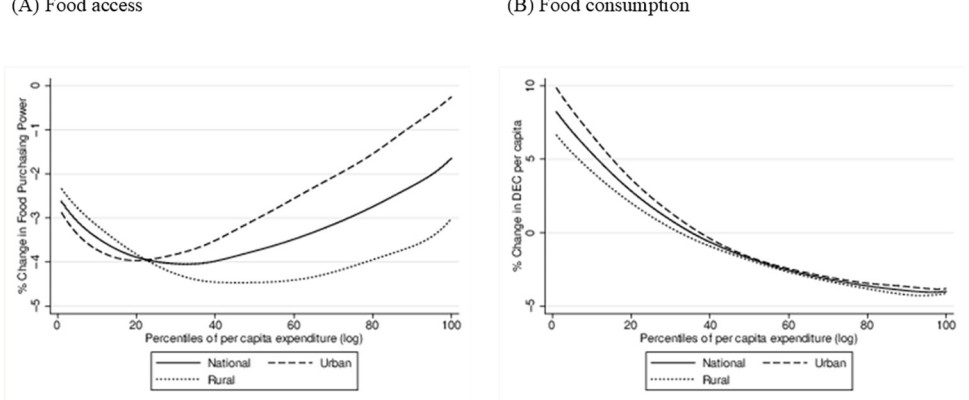

**Fig 3. Change in food access and food sufficiency dimensions of Comoros' households by per capita expenditure and by area of residence.** Source: own elaboration.

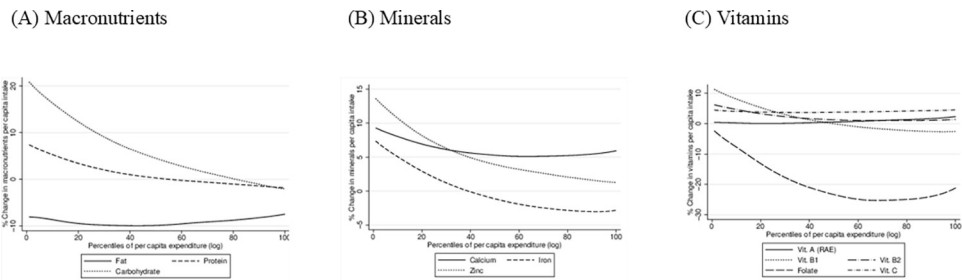

**Fig 4. Change (%) in daily macronutrients (fats, proteins and carbohydrates), minerals (calcium, iron, zinc) and vitamins intakes per capita of Comoros' households.** Source: own elaboration.

on average, within rural populations and among households below the 50th percentile of per capita expenditure overall. In rural areas, the deterioration in food access peaks in the middle of the distribution. In this context, rural dwellers may face greater challenges due to higher costs of fertilizers and other agricultural commodities, as well as transportation, making food access even more costly. In urban areas, households below the 20th percentile are more affected, experiencing an average food access loss of 4%.

Regarding the **food consumption** dimension (Fig 3, panel B), measured by the change in **DEC per capita** at the household level, we observe similar patterns within rural and urban populations. Notably, there is a consistent reduction in caloric intake among wealthier households while in poorer households it increases in both areas. This may be related to the fact that the Russia-Ukraine conflict impacted primarily the prices of imported and expensive goods like organ meats and nuts, thus negatively affecting the diet of wealthier households. Conversely, the prices of more affordable food products like red meat, sweets and oils and fats were reduced in real terms, and consumption increased (See Fig 2). These food products are also the main contributors to caloric intake in Comoros (See Table 2), thus resulting in an increased daily energy consumption (DEC) for poorer households.

We also calculated the percentage of households meeting the threshold of 2,250 kcal per capita per day before and after one year from the beginning of the Russia-Ukraine conflict. The probability of achieving **food sufficiency**, measured in terms of daily energy intake per capita, declines from 42% to 39% at the national level, from 39% to 36% in rural areas, and from 47% to 44% in urban areas between 2022 and 2023.

The impact on the **food adequacy** dimension, measured by the change in **macronutrient intakes** (Fig 4, panel A), reveals a reduction in fats intake across all households, irrespective of their per capita income. This decline in daily fat intake can be attributed to the sharp reduction in fat-rich fruits (coconut and avocado) included in the other fruits food group, as well as a decrease in the consumption of legumes, nuts & seeds. While the decrease in fat consumption may be viewed positively in a population with the nutritional profile of Comoros and other SIDS, it is crucial to delve into the type of fats introduced by this shift, as the quality of fats contributes significantly to a healthy dietary pattern [46]. Moreover, it is important to understand that this result is driven by changes in relative prices. The war has affected more the price of imported food products which in normal times are cheaper but with less nutritional value. The substitution effect would direct households to consume more local products with better nutritional value at the cost of affecting food sufficiency.

Protein intake increased for most households, primarily due to heightened consumption of flesh meats and dairy products. To interpret this result effectively in terms of nutrition, it is essential to have more information on the types of flesh meat and dairy products consumed,

along with their degree of processing. In many SIDS, flesh meats are imported and highly processed [47]. Households in the highest percentiles of per capita expenditure distribution experienced a slight decrease in protein intake, possibly reflecting a decline in their consumption of organ meats, a consequence of the reduction in the quantities of imported organ meats that had already decreased from 10,005 tons in 2021 to 412 tons in 2022 [48].

Carbohydrate consumption also increased among most households during the study period, especially among the poorest ones, with a more pronounced rise. This aligns with the sweets and white tubers increased consumption (10% and 5% respectively). The transition to foods rich in carbohydrates but poor in other nutrients is consistent with the response to degrading food access, where individuals compromise on the nutritional quality of consumed food. Moreover, by June 2022, the Comoros government implemented a subsidy program targeting bakeries to mitigate the pass-through from wheat to bread prices [49], which may be associated with the overall increase in sweets consumption.

The observed reduction in fat intakes and the increase in carbohydrate consumption resulted in a shift toward more balanced diets in terms of macronutrients. We calculated the adequacy of the caloric contribution of each type of macronutrient before and after one year of the Russia-Ukraine conflict. Initially, only 11% of households at the national level displayed a balanced diet in terms of macronutrients, but after one year of the conflict, this percentage increased to 16% (See Table A.3 in the S1 File). However, the percentage of households that do not achieve the recommended thresholds for all macronutrients remains high (32%), posing a significant concern.

The impact on **food adequacy** is also measured in terms of the change in micronutrient consumption (minerals and vitamins). The **daily mineral intakes per capita** (Fig 4, panel B) show that iron intake suffered a negative impact for wealthier households (-5% loss in daily iron intake), likely linked to the reduction in the consumption of organ meat associated with reduced imports as already mentioned. A similar impact is observed for the change in zinc per capita intakes. However, calcium intakes increased between 5% and 10%, with no remarkable difference across percentiles of per capita expenditure, aligning with the 4% increase in milk and milk products consumption estimated in Fig 2. Concerning the changes in **daily vitamin intake per capita** (Fig 4, panel C), among the studied vitamins, folate exhibits a more pronounced negative impact, consistent with the reduction in the main sources of folate in the Comorian diet (organ meat, vitamin A-rich fruits, cereals, legumes, nuts, and seeds). This reduction is more pronounced for the wealthiest households, with a -23% loss compared to the poorest households (-8%), explained by an economic restriction to more expensive food items rich in folates. For the remaining vitamins, the mean daily per capita intake shows a slight positive impact, though insufficient to overcome the level of vitamin deficiencies detected in the population. This emphasizes the need to promote local fresh fruits and vegetables production and consumption.

## 4. Discussion and conclusion

The nutritional profile of Comoros closely mirrors that of other Small Island Developing States (SIDS) in the Pacific and Caribbean, marked by high-fat diets, widespread micronutrient deficiencies, and a dual burden of malnutrition—ranging from childhood undernutrition to rising rates of anemia among women and obesity among adults and adolescents.

The ongoing Russia-Ukraine conflict has exacerbated this situation by driving up international prices for energy, transportation, agricultural commodities, and food, heavily impacting countries like Comoros that rely on imports. The surge in domestic food prices has led to a significant decline in the consumption of essential food groups such as cereals, organ meats,

legumes, nuts, and fruits. As a result, Comorian households have seen a reduction in daily energy and key nutrient intake, leaving an additional 5,010 households (3.3% of all households) with insufficient energy intake.

This conflict-induced rise in food prices has particularly affected fat intake, as the cost of fat-rich foods like coconut, avocado, nuts, and seeds has surged. Meanwhile, cereals, which are vital to Comorian diets, have also seen a decrease in consumption. Though carbohydrate intake has increased—driven by higher consumption of sweets and white tubers—the overall quality of nutrition has diminished.

In terms of micronutrients, there has been a notable reduction in iron intake among wealthier households, linked to a 17% drop in the consumption of organ meats, typically consumed by these households. Folate intake has also decreased across all households. Given the high prevalence of anemia in women of reproductive age in Comoros and other SIDS, these reductions are particularly concerning. While modest increases in other minerals and vitamins, likely due to higher consumption of flesh meats and milk, have been observed, they are insufficient to address the widespread nutritional deficiencies.

Reversing this trend and reincorporating fruits and vegetables—rich in micronutrients and low in calories—into the diets of Comorians and other SIDS populations is critical. Encouraging local production of these foods is essential to reduce dependence on global markets and ensure access to fresh, nutrient-dense foods. This aligns with the recommendations in the Global Action Programme on Food Security and Nutrition in SIDS and the Comoros Pact for Food and Agriculture [24]. Additionally, recent reviews have highlighted the lack of comprehensive nutritional policies in SIDS and called for a broadening of interventions beyond education to include food supply chains and cross-sectoral health-related initiatives [5, 9]. Our findings support this view, as global economic shocks like the Russia-Ukraine conflict can directly affect both macronutrient and micronutrient consumption, which in turn impacts health outcomes.

To address the challenges posed by food price shocks in SIDS, programs must be tailored to support the most vulnerable households. Direct cash transfers combined with nutrition education programs would enable families to afford essential food items without compromising their well-being [50]. Food vouchers redeemable for fresh fruits and vegetables, along with subsidies for locally produced foods, could further enhance food security while supporting local agricultural markets. Additionally, investments in food storage infrastructure are crucial to ensure consistent access to fresh, healthy food.

Despite the valuable insights provided, this study has certain limitations. The methodology focuses on the direct effects of price changes on consumption quantities, without considering the influence of income or substitution between food types. Moreover, the elasticities used are derived from World Bank estimates rather than calculated specifically for this study (see S1 File). Nonetheless, while these factors might affect the exact figures, the general trends in price and consumption patterns remain consistent with previous findings. Therefore, our results provide critical guidance for policymakers aiming to mitigate the impact of external shocks on the nutrition and health of populations in SIDS.

## Supporting information

**S1 File. Appendix.**
(DOCX)

**S2 File. Read me file for Comoros nutrients composition table.**
(TXT)

**S3 File. Comoros nutrients composition table.**
(XLSX)

**S4 File. Stata code and data zip file.**
(ZIP)

## Author Contributions

**Conceptualization:** Estefanía Custodio, Maria Priscila Ramos, Sofía Jimenez, Francis Mulangu, Nicolas Depetris-Chauvin.

**Data curation:** Estefanía Custodio, Maria Priscila Ramos, Sofía Jimenez.

**Formal analysis:** Estefanía Custodio, Maria Priscila Ramos.

**Investigation:** Maria Priscila Ramos, Sofía Jimenez, Francis Mulangu, Nicolas Depetris-Chauvin.

**Methodology:** Estefanía Custodio, Maria Priscila Ramos, Sofía Jimenez, Francis Mulangu, Nicolas Depetris-Chauvin.

**Project administration:** Francis Mulangu, Nicolas Depetris-Chauvin.

**Supervision:** Francis Mulangu, Nicolas Depetris-Chauvin.

**Validation:** Francis Mulangu.

**Writing – original draft:** Estefanía Custodio, Maria Priscila Ramos, Sofía Jimenez.

**Writing – review & editing:** Estefanía Custodio, Maria Priscila Ramos, Sofía Jimenez, Francis Mulangu, Nicolas Depetris-Chauvin.

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
