## [Decision Letter · Decision Letter 0]

14 Aug 2024

PONE-D-24-27344Food security and nutritional vulnerability in small island developing states: Analyzing diet adequacy and the impact of Ukraine-Russia conflict in food security and nutrition indicators in ComorosPLOS ONE

Dear Dr. Depetris-Chauvin,

Thank you for submitting your manuscript to PLOS ONE. After careful consideration, we feel that it has merit but does not fully meet PLOS ONE’s publication criteria as it currently stands. Therefore, we invite you to submit a revised version of the manuscript that addresses the points raised during the review process.

**ACADEMIC EDITOR: ****Thank you for the submission.  Please read carefully the reviewers comments and revise accordingly.  Give more attention on the novelty of your research by including research gaps and your study contributions. **

We look forward to receiving your revised manuscript.

Kind regards,

Abu Hayat Md. Saiful Islam

Academic Editor

PLOS ONE

Journal Requirements:

3. In the online submission form, you indicated that the data of the Comoros National Household survey is available upon request from the Nataional Statistics Institute of Comoros (INSEED), at INSEED's website https://www.inseed-comores.org/category/enquetes/ehcvm-2020/. The Comoros Nutrients Composition Table elaborated by the authors will be held in a public repository (RepiSalud)

5. Please amend either the title on the online submission form (via Edit Submission) or the title in the manuscript so that they are identical.

Additional Editor Comments:

Manuscript ID PONE-D-24-27344 entitled "Food security and nutritional vulnerability in small island developing states: Analyzing diet adequacy and the impact of Ukraine-Russia conflict in food security and nutrition indicators in Comoros" which you submitted to PLOS ONE, has been reviewed. The comments of the reviewer(s) are included at the bottom of this letter.

The reviewer(s) have recommended some revisions to your manuscript. Therefore, I invite you to respond to the reviewer(s)' comments and accordingly revise your manuscript.

Reviewers' comments:

Reviewer's Responses to Questions

**Comments to the Author**

1. Is the manuscript technically sound, and do the data support the conclusions?

Reviewer #1: Yes

Reviewer #2: Partly

2. Has the statistical analysis been performed appropriately and rigorously? 

Reviewer #1: Yes

Reviewer #2: Yes

3. Have the authors made all data underlying the findings in their manuscript fully available?

Reviewer #1: Yes

Reviewer #2: Yes

4. Is the manuscript presented in an intelligible fashion and written in standard English?

Reviewer #1: Yes

Reviewer #2: Yes

5. Review Comments to the Author

Reviewer #1: Review on “Food security and nutritional vulnerability in small island developing states (SIDS): Analyzing diet adequacy and the impact of Ukraine-Russia conflict in food security and nutrition indicators in Comoros”

Summary

This paper examines food and nutritional security in Comoros which is one of the least studies small island developing states. Moreover, the authors examine the effects of Ukraine-Russia War on the food and nutritional security, by using simulations. They found that Comoros experiences limited access to sufficient and nutritious diets, and a decline in food access and purchasing power for all households, particularly the poorer and rural. Their contribution is to fill a crucial gap in the literature by examining the impact on nutrients, providing essential information for targeted policy recommendations. They suggest that policies promoting the consumption of fresh, nutrient-rich foods with low fat content are crucial to mitigate the malnutrition triple burden.

This paper uses solid descriptive analysis to answer their research questions.

I ask the authors to share the Appendix for a further review.

Major comments

1. Regarding the literature about Ukraine-Russian War and food security, you may want to include the following papers as past studies.

Trade disruption

Lin, F., Li, X., Jia, N., Feng, F., Huang, H., Huang, J., ... & Song, X. P. (2023). The impact of Russia-Ukraine conflict on global food security. Global Food Security, 36, 100661.

Worsen cost of agricultural input

Alexander, P., Arneth, A., Henry, R., Maire, J., Rabin, S., & Rounsevell, M. D. (2023). High energy and fertilizer prices are more damaging than food export curtailment from Ukraine and Russia for food prices, health and the environment. Nature Food, 4(1), 84-95.

2. In Equation (1), why do you add negative sign in the equation?

3. Line 216, where does ε show up in equations? How did you translate the changes in food prices into changes in food quantities, busing the estimates of price-demand elasticities?

4. How did you derive “the mean amount of fat consumed in Comoros is 91 grams per day per capita, which closely aligns with the mean fat intake in Fiji and Guam Island, countries with some of the highest obesity rates globally”? Table 1 shows the mean fat intake from 797 to 869 kcal per day per capita. I suggest that you include the value of kilogram in a table to easily understand the statement.

5. Line 360, according to Figure 3B, households below 40 % of per capita expenditure increased their food consumption. How do you interpret the result?

6. The authors argue the need to promote local fresh fruits and vegetables production and consumption. However, the results show that Ukraine-Russian war has positively affected on food adequacy except for Folate. I know the Comoros needs more improvement in food and nutrition intake but it does not sound the implication derived from the analysis. I suggest that the authors rewrite their implication. This is a surprising result for me.

7. Line 435, for example, “the conflict's effects resulted in a notable reduction in iron intake for all households” is strong conclusion because 40% of households experience increase in Iron intake. The authors need to rewrite the conclusion.

Minor comments

Line 91, delete r between average and SDG2.

Line 99, delete e between by and the.

Line 113, delete average.

Line 115, delete but.

Reviewer #2: Reviewed comments on the manuscript entitled "Food security and nutritional vulnerability in small island developing states: Analyzing diet adequacy and the impact of Ukraine-Russia conflict in food security and nutrition indicators in Comoros" submitted to the Journal of "PLOS ONE" for publication.

PONE-D-24-27344

I have gone through the manuscript carefully and with high interest. In this manuscript, the authors tried to address the Food Security situation in Comoros amidst the Russia-Ukraine conflict. As a reviewer, my comments are as follows:

General comments: I think the insights produced in this manuscript will be helpful for policy formulation in Comoros. It is a timely and demand-driven research. Without a few punctuational mistakes, I found the manuscript well-written.

1. Title: I recommend the title should be “Food Security and Nutritional Vulnerability in Comoros: The Impact of Russia-Ukraine Conflict”.

2. Affiliation: If it is possible, write the affiliation in International Language for the general readers.

3. Abstract:

The Sustainable Development Goal #2 aims to eradicate hunger. Avoid # sing.

4. Keywords: I recommend the keywords and the order is as follows-

Food security, Nutrition, Diet adequacy, Russia-Ukraine Conflict, Comoros

5. Introduction:

The introduction is a bit large. To avoid it, authors may keep some parts of it in the footnote. For instance, lines 50 to 52.

I found both Russia-Ukraine and Ukraine-Russia in the manuscript. To maintain homogeneity, I suggest Russia-Ukraine.

Lines 63 to 64 are not clear. Please, rewrite.

In line 67, change the word provision to production.

From lines 77 to 94, readers will lose readability. Keep some in the footnote.

The average r SDG2 score. Is it ok? Check.

Line 99, exacerbated byethe COVID-19 pandemic. Please, check.

Lines 103 to 104. Check.

Line 115. Check.

Line 120, change provision to production.

6. Methodology:

Sufficiently explained.

7. Results:

This part is also well explained.

8. Discussion and conclusion:

In this part, I found some redundancy. For instance, from lines 418 to 446. I am not convinced by the present discussion. Please, rewrite by avoiding redundancy.

I recommend a separate conclusion section.

The conclusion is also poor. Please rewrite.

6. PLOS authors have the option to publish the peer review history of their article (what does this mean?). If published, this will include your full peer review and any attached files.

Reviewer #1: No

Reviewer #2: **Yes: **Dr. Mohammad Saiful Islam; Bangladesh Livestock Research Institute

---

## [Author Response · Author response to Decision Letter 0]

14 Oct 2024

We have attached a letter with the detailed response to the reviewers.

---

## [Editor Report · Decision Letter 1]

23 Oct 2024

Food security and nutritional vulnerability in Comoros: the impact of Russia-Ukraine conflict.

PONE-D-24-27344R1

Dear Dr. Nicolas Depetris-Chauvin,

We’re pleased to inform you that your manuscript has been judged scientifically suitable for publication and will be formally accepted for publication once it meets all outstanding technical requirements.

Kind regards,

Abu Hayat Md. Saiful Islam

Academic Editor

PLOS ONE

Additional Editor Comments (optional):

Please adhere to the PLOS ONE style, references  and supplementary  information details.
---

## [Editor Report · Acceptance letter]

1 Nov 2024

PONE-D-24-27344R1 

PLOS ONE

Dear Dr. Depetris-Chauvin, 

I'm pleased to inform you that your manuscript has been deemed suitable for publication in PLOS ONE. Congratulations! Your manuscript is now being handed over to our production team.

Kind regards, 

on behalf of

Dr. Abu Hayat Md. Saiful Islam 

Academic Editor

PLOS ONE